# I2-Imidazoline Ligand CR4056 Improves Memory, Increases ApoE Expression and Reduces BBB Leakage in 5xFAD Mice

**DOI:** 10.3390/ijms23137320

**Published:** 2022-06-30

**Authors:** Bibiana C. Mota, Nathan Ashburner, Laura Abelleira-Hervas, Liyueyue Liu, Robertas Aleksynas, Lucio Claudio Rovati, Gianfranco Caselli, Magdalena Sastre

**Affiliations:** 1Department of Brain Sciences, Imperial College London, Hammersmith Hospital, Du Cane Road, London W12 0NN, UK; b.mota@imperial.ac.uk (B.C.M.); nathan.ashburner17@imperial.ac.uk (N.A.); l.abelleira-hervas17@imperial.ac.uk (L.A.-H.); liyueyue.liu19@imperial.ac.uk (L.L.); robertas.aleksynas19@imperial.ac.uk (R.A.); 2Rottapharm Biotech S.r.l., 20900 Monza, Italy; lucio.rovati@rottapharmbiotech.com (L.C.R.); gianfranco.caselli@rottapharmbiotech.com (G.C.)

**Keywords:** imidazoline, astrocyte, Alzheimer’s disease, amyloid-β, blood–brain barrier, aquaporin-4

## Abstract

Recent evidence suggests that I2-imidazoline ligands have neuroprotective properties in animal models of neurodegeneration, such as Alzheimer’s disease (AD). We recently demonstrated that the I2-ligand BU224 reversed memory impairments in AD transgenic mice and this effect was not because of reductions in amyloid-β (Aβ) deposition. In this study, our aim was to determine the therapeutic potential of the powerful analgesic I2-imidazoline ligand CR4056 in the 5xFAD model of AD, since this ligand has been proven to be safely tolerated in humans. Sub-chronic oral administration of CR4056 (30 mg/kg for 10 days) led to an improvement in recognition memory in 6-month-old 5xFAD mice, but not in wild-type littermates, without affecting Aβ levels or deposition. Our results also revealed a change in the profile of microglia by CR4056, resulting in a suppression of pro-inflammatory activated microglia, but increased the density of astrocytes and the expression of ApoE, which is mainly produced by these glial cells. In addition, CR4056 restored fibrinogen extravasation, affecting the distribution of markers of astrocytic end feet in blood vessels. Therefore, these results suggest that CR4056 protects against Aβ-mediated neuroinflammation and vascular damage, and offers therapeutic potential at any stage of AD.

## 1. Introduction

The cardinal symptoms of Alzheimer’s disease (AD) in the early stages are memory loss and modest changes in behaviour. These symptoms slowly progress with additional deterioration of the patient’s executive function, language and visuospatial abilities, with the patient gradually declining into total dementia and losing their autonomy [1]. The average survival time from the point of diagnosis is 5.7 years for women and 4.2 years for men [2]. Unfortunately, current treatments for AD only provide short-term symptomatic relief rather than disease-modifying benefits. These include acetyl cholinesterase inhibitors (AchEI), such as donepezil, rivastigmine and galantamine and the *N*-methyl-D-aspartate (NMDA) receptor antagonist memantine [3] (Figure 1A). These all work from mild to severe AD, showing benefits in cognitive and behavioural symptoms. The only approved NMDA antagonist, memantine, aims to reduce the prolonged release of glutamate in order to reduce NMDA hyperactivation. NMDA receptors overreaction may lead to neurodegeneration and loss of synaptic function through chronic excitotoxicity [4]. Memantine can be administered with an AChEI since they have a complementary mechanism of action, for a combined improved cognitive effect [5].

With current treatments lacking curative effects and producing unwanted side effects, the development of new disease-modifying interventions of AD is imperative. There have been various clinical trials targeting the well-established roots of AD of tau and amyloid formation, but with little success. Aβ immunotherapy, such as aducanumab, has been approved without controversies by the FDA and various targets of the tau pathway including kinase inhibitors have shown poor therapeutic effects [6,7]. Other treatments targeting Aβ generation, such as secretase inhibitors, have lacked clinical effect, with γ-secretase inhibitors failing to improve cognitive status and even showing concerning clinical complications, while β-secretase cleaving enzyme-1 (BACE1) inhibitors were poorly tolerated by patients, with no cognitive improvements [8,9]. With these approaches providing little breakthrough, investigation into alternative therapeutic targets of AD is paramount.

A new area that is emerging is targeting neuroinflammation and blood–brain barrier (BBB) damage [10,11,12,13]. Aβ has been reported to activate glial cells, including microglia and astrocytes, resulting in an exaggerated inflammatory response [10,12]. When microglia are activated, they migrate to the plaque and are able to phagocytose and degrade Aβ [14,15,16]. This occurs during the acute inflammatory response, which is eventually neuroprotective [17]. However, chronic inflammation is detrimental due to sustained activation of the microglia, changing their morphology and accompanied with the release of neurotoxic cytokines such as TNFα, IL-6 and IL-1β, reflecting a change in their phenotype, becoming reactive [10,12]. This pro-inflammatory response has been reported to lead to neuronal dysfunction and death, as well as an increase in Aβ generation [18,19]. Inflammatory events in AD have also been linked with increased BBB permeability [20,21,22], allowing the entrance of peripheral immune cells and molecules in the brain [13], contributing further to the pathogenesis of AD.

In this regard, recent studies with imidazoline receptor ligands have highlighted their potential role as new therapy for AD, because of their anti-inflammatory and anti-apoptotic effects [23]. These receptors were originally identified in the early 1990s, as certain α2-adrenoreceptor ligands such as clonidine and idazoxan were found to target a distinct, novel type of receptor [23]. Specifically, the I2 receptor (I2IR) subtype is of therapeutic interest in neurodegeneration, since it has great expression in the brain and anti-inflammatory properties [24]. Interestingly, I2IRs have been found increased in the brains of AD patients [25,26]. I2IRs are mainly expressed in the outer layer of mitochondria in astrocytes and are associated with centrally mediated effects, such as anti-nociception, neuroprotection and increased expression of Glial fibrillary acidic protein (GFAP) [27,28,29,30,31]. Due to the neuroprotective function of the I2 receptor, ligands have been designed to have high affinity for I2IRs and low for α2 receptors, including BU224, LSL60101 and CR4056 [32,33] (Figure 1B), which have also been shown to readily cross the BBB [34]. The pharmacology of I2IRs is more complex than that of most conventional receptors. Indeed, single agents such as idazoxan could behave as putative antagonist towards most of I2 triggered effects, still maintaining some “agonist activity” on others, as discussed, for instance, by Vellani et al. [35]. Therefore, a conventional agonist/antagonist pharmacology on I2 receptors is difficult to perform. Having said this, CR4056 was demonstrated to be specific for I2IRs, being devoid of activity/affinity towards a large series of cross-targets [33]. Consequently, most of its activities are antagonized by the purported I2 antagonist idazoxan but not by other antagonists towards different targets, such as I1, adrenergic α2 and α1, or opiate receptors [31,33]. Interestingly, CR4056 has been utilised in a clinical trial for pain and was well tolerated by patients [36]. Because of its proven safety profile in humans [36], CR4056 seems to have promising prospects in future clinical trials. 

A recent publication from our lab [37] has demonstrated that sub-chronic treatment with the highly selective I2IR ligand BU224 significantly improved cognitive impairment in the 5xFAD mouse model of amyloidosis, at stages with high amyloid deposition and neuronal loss. This cognitive amelioration was only observed in transgenic mice and not in wild-type (WT) animals, suggesting that the beneficial effects were secondary to Aβ deposition. In addition, treatment with BU224 did not affect amyloid pathology or apoptosis markers and, therefore, the therapeutic effects were potentially associated with reduced neuroinflammation, increased astrocytic density and decreased NMDA-mediated toxicity [37]. Other groups have also observed the beneficial effects of other I2 ligands in AD models, with different outcomes regarding Aβ deposition and astrocytic proliferation [38,39], suggesting that they may affect Aβ levels depending on the drug structure and the dose and duration of the treatment.

In this study, our aim was to determine the therapeutic potential of the powerful analgesic I2-imidazoline ligand CR4056 [35,36] in the 5xFAD model of AD, since this ligand has been proven to be safely tolerated in humans. It was recently reported that CR4056 has anti-inflammatory properties [35,40] and thus we sought to determine whether treatment with this ligand affected the activation profile of microglia, by assessing changes in the morphology and phagocytic profile in 5xFAD mice. In addition, we further explored the potential beneficial effects of I2IR ligands regulating functions controlled by astrocytes, such as BBB maintenance. Our results suggest that sub-chronic treatment with CR4056 reduces memory deficits and targets inflammatory pathways, decreasing the expression of pro-inflammatory cytokines, as well as improving BBB functionality in the 5xFAD model of AD.

## 2. Results

### 2.1. CR4056 Reverses Memory Deficits in the 5xFAD Model of AD

We treated 5xFAD mice and WT littermates at 6 months of age, when 5xFAD animals have extensive amyloid deposition and memory loss [37] with 30 mg/kg CR4056 for 10 days (Figure 2A). To determine the effects of CR4056 treatment on behaviour, mice underwent open field, Novel Object recognition (NOR) and Object location task (OLT) tests. Six-month-old WT and 5xFAD mice treated with CR4056 did not show any motor deficits or altered anxiety, measured by the open field paradigm, without changes in distance moved or thigmotaxis (Figure 2B,C).

In addition, and as reported in our previous study using BU224 [37], 5xFAD mice displayed recognition and spatial memory impairments at 6 months of age, determined by the NOR and the OLT, showing no preference for the new object or the displaced object, while WT mice did not display any alterations (Figure 2E,G). Interestingly, CR4056 reversed the object recognition deficits in the transgenic mice, with values similar to WT animals (Figure 2E). 

WT mice did not show improvements with the treatment, indicating CR4056 affected mechanisms altered as a consequence of Aβ deposition.

### 2.2. Sub-Chronic Administration of CR4056 Does Not Affect Aβ Deposition

We next examined whether CR4056 treatment affected Aβ deposition and distribution by immunohistochemical staining with Thioflavin-S and antibody 6C3. As reported with BU224 treatment, CR4056 did not cause any changes either in amyloid-β plaque burden or number of plaques by Thio-S staining in cortex and hippocampus of 5xFAD mice (Figure 3A–E). In addition, total amyloid burden measured by 6C3 staining did not reveal alterations in Aβ deposition (Figure 3F–H). 

Consistent with the immunostaining results, the levels of Aβ were found not altered by CR4056 treatment in homogenates of cortex and hippocampus, measured either by ELISA (Figure 3I,J) or by Western blot with 6E10 (Figure 3K,L,Q). Furthermore, the examination of the effect of CR4056 on the mechanisms of generation of Aβ (Figure 3N–Q) demonstrated that treatment with CR4056 did not affect the expression of full length amyloid precursor protein (APP) (Figure 3O,P) and the levels of the carboxy-terminus fragments (CTFs) in the cortex and hippocampus (Figure 3M,N). These results suggest that CR4056 did not have any effect on the cleavage or the expression of APP.

### 2.3. Effect of CR4056 on Astrocyte Density

As reported previously, I2-imidazoline ligands increase GFAP expression [30,41] and the number of astrocytes [37] in rodents. We therefore measured the GFAP-positive area in the cortex and hippocampus of 5xFAD treated with CR4056 (Figure 4A–E). In agreement with our previous results using BU224, GFAP staining was increased with CR4056 administration, particularly in the cortex of 5xFAD mice (Figure 4A,C). In line with this, Western blot with the astrocytic marker glutamine synthase (GS) revealed a 44% increase in the levels of this marker in the cortex of 5xFAD mice treated with the imidazoline ligand (Figure 4F,L).

Interestingly, we observed a significant increase in the expression of proteins that are synthesized by glial cells [16], such as Apolipoprotein-E (Apo-E) (Figure 4H,L), which could be potentially related to the increase in the density of astrocytes by I2-imidazoline ligands, since this protein is mainly produced in these astroglial cells [16]. In addition, the insulin-degrading enzyme (IDE), responsible for the degradation of Aβ in cortex and hippocampus of 5xFAD mice and mainly synthesized by glial cells, was also elevated in cortex and hippocampus, after CR4056 compared with vehicle-treated controls (Figure 4J–L).

### 2.4. Anti-Inflammatory Effect of CR4056 in 5xFAD Mice

We and others have shown previously the anti-inflammatory effects of I2-imidazoline ligands, by reducing the density of microglia and pro-inflammatory cytokines [35,37,40]. We therefore sought to determine the effect of CR4056 on the density and activation state of microglia, by performing analysis of the number and morphological changes in microglia by staining with Iba-1 and CD68 (a marker for phagocytic microglia). 

5xFAD mice treated with CR4056 showed reduced Iba-1 positive coverage in the cortex compared with vehicle-treated mice (Figure 5A,B). To examine alterations in microglia phenotype, we carried out the staining with CD68, in order to identify changes in phagocytic microglia, showing no changes in area covered or in number of cells (Figure 5F–J), suggesting that the treatment led to a reduction in classically activated microglia. Furthermore, CD68 positive microglia displayed enlarged soma and retracted processes in 5xFAD mice (Figure 5F), and sometimes rod-like morphology (see example of amplification of microglia in hippocampus, Figure 5F), typical of a pro-inflammatory phenotype, which was reversed after treatment with CR4056. Analysis of microglia morphology using HALO software (Figure 4K) suggested increases in process length and area in microglia of CR4056-treated mice compared with control animals particularly in cortex (Figure 5K–N).

Additionally, the expression of the pro-inflammatory cytokines Il1-β, IL-6 and TNFα was measured by ELISA in the cortex and hippocampus of 5xFAD mice (Figure 5O–T). Our results show that CR4056 treatment led to a reduction in the levels of TNFα in the cortex and hippocampus of 5xFAD mice (Figure 5Q,R), without significant changes in the other cytokines.

In summary, the results showed that CR4056 has anti-inflammatory effects in 5xFAD mice, affecting the activation state of microglia and their number.

### 2.5. CR4056 Reverses BBB Leakage in 5xFAD Mice

Because astrocytes are an important component of the BBB, and imidazoline ligands have been reported to have cardiovascular effects [23], we assessed the levels extravasation of fibrinogen from blood vessels, as a measurement of BBB leakage. Our results show that 5xFAD mice display increased fibrinogen staining in the cortex, compared with WT mice (Figure 6A,B). Interestingly, treatment with CR4056 reduced the extravasation of fibrinogen in the cortex of 5xFAD mice (Figure 6B).

The main marker of the astrocytic end feet is the aquaporin-4 (AQP4) water channel, which is involved in the clearance of waste, cerebrospinal fluid (CSF) homeostasis and other physiological functions [42]. Because I2-imidazoline ligands affect markers of astrocytes, we determined whether CR4056 altered the expression of AQP4 and the potential changes in their distribution along the blood vessels. We have previously reported that 5xFAD mice showed a reduced distribution of AQP4 in blood vessels [13]. Interestingly, we observed a reversal of this effect in animals treated with CR4056, showing co-localization of AQP4 with the marker of endothelial cells CD10 in animals treated with CR4056 (Figure 6D). These changes were not due to alterations in the total expression of AQP4, which was analysed by Western blot in the cortex and hippocampus of 5xFAD mice (Figure 6E–G).

## 3. Discussion

The lack of effective therapies for AD, with recent failings of amyloid-based clinical trials, suggests that disease-modifying treatments are only useful as preventive therapies, highlighting the importance of investigating new therapeutic avenues, in particular for use at late stages of the disease. The present study illustrates the beneficial effects of CR4056, a potent I2-ligand, on neuroinflammation and BBB functionality, in a mouse model of 5XFAD, at an age that presents heavy amyloid load, neuronal and memory loss. Interestingly, this drug has been used in clinical trials to treat pain [36], without side effects, suggesting that it could be used to treat AD patients without having to explore its safety in humans.

The beneficial effects of CR4056 on spatial memory do not seem to be a consequence of changes in Aβ deposition, in agreement with our recent report using BU224 in the same transgenic line [37], demonstrating no changes in APP processing. This is in contrast with the results of Vasilopoulou et al. using LSL60101 [39], which showed reductions in Aβ deposition, which could be related to a longer administration. Studies using a different strategy, by treatment with the endogenous imidazoline-I1 ligand agmatine in mice injected with synthetic Aβ, also reported reductions in Aβ levels, suggesting perhaps that these drugs may affect mechanisms of amyloid degradation or clearance [43]. In line with this, we found that CR4056 increased the expression of IDE, and others have reported elevated levels of neprilysin with the I2-ligand MCR5 in the in SAMP8 mouse model of senescence [44]. However, IDE has many other functions, besides being involved in the degradation of Aβ. In fact, it is involved in the control of glucose (by affecting the degradation of insulin); it behaves as a heat shock protein and regulates the ubiquitin–proteasome system, suggesting a major implication in protein turnover and cell homeostasis [45].

I2IRs are located mostly in glial cells and we and others have described the anti-inflammatory effects of I2-imidazoline ligands. Although there is evidence that I2IR drugs reduce the density of microglia in different models of CNS neuroinflammation [35,37,39] as well as the levels of pro-inflammatory cytokines and ROS [37,44,46,47], no studies have so far investigated the effect of imidazoline ligands on microglia phenotype, including morphological changes. We present now in our study that CR4056 affects also the microglia phagocytic profile, by using the marker CD68. In addition, the morphological changes detected in our model suggest that CR4056 reduces the activation state of a particular subset of microglia, which must be pro-inflammatory. Because the 5xFAD model has extensive Aβ deposition at this age, it is challenging to perform morphological analysis, showing great variability of results, since the phenotype of microglia is different depending on their localization, whether they are close to the amyloid plaque or away from them.

The potential effects of I2-imidazoline drugs on the density of astrocytes have been proven controversial, with some studies showing an increase in astrocytes and others a reduction. The reports published by our group, either with BU224 or CR4056, showed an increase in astrocytic markers GFAP and GS by imidazoline ligands, in agreement with previous studies in rats [30,37,41]. However, it is possible that the length of the treatment may affect these results and longer times may result in a different outcome. Interestingly, in our study, proteins produced by astrocytes such as ApoE were found to be increased by CR4056 treatment. Observations from our lab indicate that ablation of astrocytes in models of AD results in a reduction in ApoE [48,49] and IDE secretion [49], indeed suggesting that the astrocytes contribute to enhanced production of ApoE and IDE, as pointed out in the present study.

ApoE is a critical protein in AD, because it has multiple functions, including amyloid clearance, regulation of the microglia response to amyloid plaques, effects on neurite outgrowth and BBB function by binding to LPR receptors [50]. In line with this, in the present study, we show that sub-chronic treatment with CR4056 reduced BBB leakage, by detection of fibrinogen extravasation, and affected the distribution of AQP4. Interestingly, it was previously published that imidazoline receptor ligands protect against cardiovascular dysfunction [51]; in particular, treatment with the I2IR ligand 2-BFI in a model of stroke, the middle cerebral artery occlusion model in rat, provided strong neuroprotection and protected the integrity of the cerebral vasculature, using injections with FITC-dextran [52]. In future studies, it would be interesting to determine the mechanism involved in these changes in blood vessel integrity, and examine whether this is due to an anti-inflammatory effect or a direct effect of I2IRs on components of the BBB, such as tight junction regulation in endothelial cells. Interestingly, vascular smooth muscle and endothelial cells express I2IRs [53]. The reversal of neurovascular dysfunction in AD is a promising therapeutic property of this drug, which we have also observed recently by administration of another anti-inflammatory molecule, Annexin-A1 (ANXA1) [13].

Overall, CR4056 offers a great therapeutic potential for AD patients, at any stage of the disease, even when they have extensive amyloid deposition and neuronal loss, by affecting the activation of glial cells and reverting BBB breakdown, which have been proven to accelerate disease progression.

## 4. Materials and Methods

### 4.1. Materials and Antibodies

The following antibodies were utilised in our study: 6E10 (against Aβ1–16) from Covance; MOAB-2 clone 6C3 against Aβ from Merck-Millipore (Burlington, MA, USA); anti-apolipoprotein E (ApoE) and Aquaporin-4 (AQP-4) from Santa Cruz; anti-insulin-degrading enzyme (IDE) and anti-β-actin from Abcam; anti-ionized calcium-binding adaptor molecule 1 (Iba1) from Wako; anti-CD68 from Biolegend; Rat anti-GFAP (clone 2.2B10) from Invitrogen or from DAKO; anti-fibrinogen from DAKO and anti-CD31 from BD Biosciences. All other reagents were purchased from Invitrogen or Sigma, unless otherwise indicated. 

### 4.2. Animals and Treatment

Six-month-old female 5xFAD mice (*n* = 38) and WT C57Bl6 matched controls (*n* = 37) were utilised for the study, in line with the UK Animals (Scientific Procedures) Act 1986 and ethical standards outlined by the Imperial College London Animal Welfare and Ethical Review Body. The 5xFAD model overexpresses the human APP gene (APP695) with Florida (I716V), Swedish (K670N/M671L), and London FAD mutations (V717I), in addition to human Presenilin1 (PS1) with M146L and L286V FAD mutations (54). The 5xFAD model presents as a severe form of AD, showing amyloid deposition already at 1.5 months of age and glial activation by 2 months. Cognitive deficits and neuronal loss are detectable at 4–5 and 6 months, respectively [54]. Hence, 6 months of age was appropriately chosen in order to investigate animals with severe neuropathology and cognitive deficits, as in our previous studies [37]. CR4056 was administered at 30 mg/kg by gavage once per day for 10 days, based on previous reports [33,55] (Figure 2A).

### 4.3. Open Field (OF)

Mice were allowed to freely explore a 45 by 45 cm arena for 5 min, and thigmotaxis, velocity, and total distance moved were assessed using Ethovision XT software (Noldus).

### 4.4. Object-Location Test (OLT)

Hippocampal-dependent spatial memory of the mice was tested using the OLT task. During the training phase, mice were allowed to explore two identical objects made from large lego bricks for 10 min in a circular arena (45 cm diameter) with intra-maze cues (distinguishable patterns and drawings on the vertical walls of the arena) and were subsequently returned to their cages. Twenty-four hours later, one of the objects was moved to a novel position in the arena and the mouse was once again allowed to explore the two objects for 10 min (testing phase). The total time the mouse spent actively exploring each object (sniffing/chewing, during both training and testing phases) was recorded using EthoVision XT software (Noldus). The time spent exploring each object was then calculated as a percent of total object exploration.

### 4.5. Novel Object-Recognition Test (NOR)

Recognition memory of the mice was tested using the NOR test. During training, mice were allowed to explore two identical objects made from large lego bricks (different objects to those used in the OLT task) for 10 min in a maze similar to that described in the OLT section (see previous section). Mice were then returned to their cages and 24 h later, one of the two identical objects was replaced with an object of different shape and colour. The mice were returned to the arena and allowed to explore the two non-identical objects for 10 min (testing trial). The percent of total object exploration was calculated as previously described in the OLT section.

### 4.6. Western Blotting

Brain homogenates were extracted from the frontal cortex and the hippocampus of 5xFAD mice with radioimmunoprecipitation (RIPA) buffer (1% Triton X-100, 1% sodium deoxycholate, 0.1% SDS, 150 mm NaCl and 50 mm Tris–HCl; pH 7.2) supplemented with Roche complete protease inhibitor. Following this, extracts were centrifuged for 10 min at 13,000 rpm at 4 °C and supernatants containing the proteins were collected. Protein concentration was determined using the Bradford assay.

Equal amounts of protein were boiled in NuPAGE 4x LDS buffer at 95 °C for 5 min. The samples were loaded in a 4–12% NuPAGE gel in MES buffer at 80 V for 20 min and then at 120 V for 40 min. The gels were then transferred to PVDF membranes at constant 400 mA for 1 h. The membranes were then blocked with 5% non-fat milk in TBST (8.8 g NaCl, 100 mL Tris HCI pH 8, 500 μL Tween-20, up to 1 L in dH2O) for 1 h to avoid binding between gel and antibody. The membranes were then washed 3 times with TBST 3 times for 10 min. Afterwards, membranes were incubated with primary antibodies (1:1000) at 4 °C overnight, with the exception of β-actin (1:5000 for 1 h at room temperature). Following additional washing with TBST for 3 times, the membranes were incubated with secondary antibodies in blocking solution (5% non-fat milk in TBST) for 1 hr at room temperature. Finally, after further washing, the membranes were developed using ECL in a GENEGNOME device. For re-blotting, Re-Blot Plus Strong Solution (Millipore) was utilised and the staining procedure was repeated. ImageJ was used to analyse the results, with β-actin used to normalize the data. 

### 4.7. Immunohistochemistry

To visualise the distribution and morphology of astrocyte and microglia, staining of floating sections was performed using GFAP and Iba1 antibodies, respectively. MOAB-2, clone 6C3 was utilised as a pan-Aβ antibody to assess the percentage of area covered of amyloid. Fibrinogen staining was used to determine BBB leakage; 30μm sections were obtained by using a cryostat and were treated briefly for 20 min with 0.6% H_2_O_2_ in TBS. The sections were then permeabilized in TBS Triton X-100 0.25% (TBS-Tx) for 30 min and subsequently blocked for 1hr with 10% FBS in TBS-Tx 0.1%. Antigen retrieval followed by permeabilization with 98% formic acid for 5 min was performed as an additional phase for Aβ staining. Sections were incubated with primary antibodies overnight at 4 °C, including anti-Aβ MOAB-2 (6C3) (Millipore) at 1:1000; and anti-GFAP (Invitrogen, Waltham, MA, USA) at 1:500, anti-Iba1 (Wako) at 1:500; anti-fibrinogen (Dako) at 1:2000; anti-AQP4 (Santa Cruz) at 1:100 or 1:200 in 2% NHS in TBS-Tx 0.02% overnight. The following day, the sections were washed and incubated with the appropriate secondary antibody at 1:500 in 5% FBS in TBS-Tx 0.1% for 2 h at room temperature. Sections were then enhanced with avidin–biotin complex (ABC, Vector Labs, Newark, CA, USA), and subsequently placed in diaminobenzidine (DAB, Sigma, Burlington, MA, USA) to visualise the staining. The sections were then sequentially washed in ddH_2_O then PBS and were mounted on slides and left to dry overnight. The next day the slides were dehydrated using increased concentrations of ethanol and then in xylene and mounted in Permount. For immunofluorescence co-staining of AQP4 and CD31, fluorescent secondary antibodies (1:400 Alexa Fluor, Invitrogen) were used and slides were mounted with ProLong Gold (Invitrogen) and imaged with a confocal microscope.

### 4.8. Thioflavin-S Staining

Thioflavin-S was used to visualise aggregated amyloid. For Thioflavin-S staining, sections were incubated with 1% Thioflavin-S for 7 min. The sections were then differentiated twice in 70% ethanol and rehydrated with water. Following this, they were placed in PBS for 10 min and mounted on POLYSINE slides with ProLong Gold antifade (Invitrogen). Thioflavin-S-stained sections were examined using HWF-1 Zeiss fluorescence microscope in 470 nm wavelength. 

### 4.9. ELISA

Brain homogenates from the frontal cortex were analysed for Aβ and pro-inflammatory cytokines by ELISAs. The Aβ_1–42_ levels were measured using High-Sensitivity Human Amyloid β42 ELISA kits from Millipore (Merck). Inflammatory cytokines IL-1β, IL-6 and TNFα were analysed by the ELISA development kits (Peprotech, Cranbury, NJ, USA), according to the manufacturer’s instructions. 

### 4.10. Analysis of Images

ImageJ software was utilised to analyse the percentage of area coverage of both ThioS- and 6C3-stained plaques in the cortex and hippocampus (2–3 sections per animal and 2–7 sections per animal, respectively). ImageJ was also utilised to count the number of ThioS-stained plaques. This was performed by converting the images to 16-bit greyscale and choosing an appropriate threshold. 

The percentage of area coverage and number of cells stained with GFAP, Iba1 and CD68 in the cortex and hippocampus were analysed using HALO software (3–5 sections per animal). The morphology of CD68-stained cells was also assessed with HALO, with process length and area being analysed.

### 4.11. Statistics

Data were tested for equal variance and normal distribution. All the data are expressed as means ± SEM. The detection of outliers was conducted by Grubbs’ test. One-way ANOVAs, two-way ANOVAs with Tukey’s post hoc analysis or a one-tailed Student’s *t*-test using GraphPad Prism Version 7.0 software were used for statistical analysis depending on the dataset. *p* < 0.05 was the threshold for statistical significance. 

## Figures and Tables

**Figure 1 ijms-23-07320-f001:**
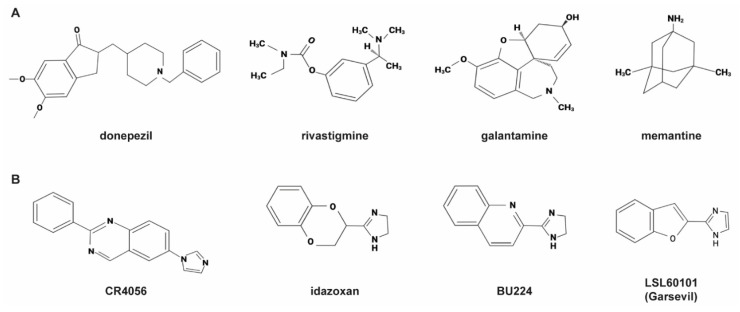
Chemical structure of drugs currently used to treat AD patients (**A**) and I2-imidazoline ligands (**B**). Created with BioRender.com.

**Figure 2 ijms-23-07320-f002:**
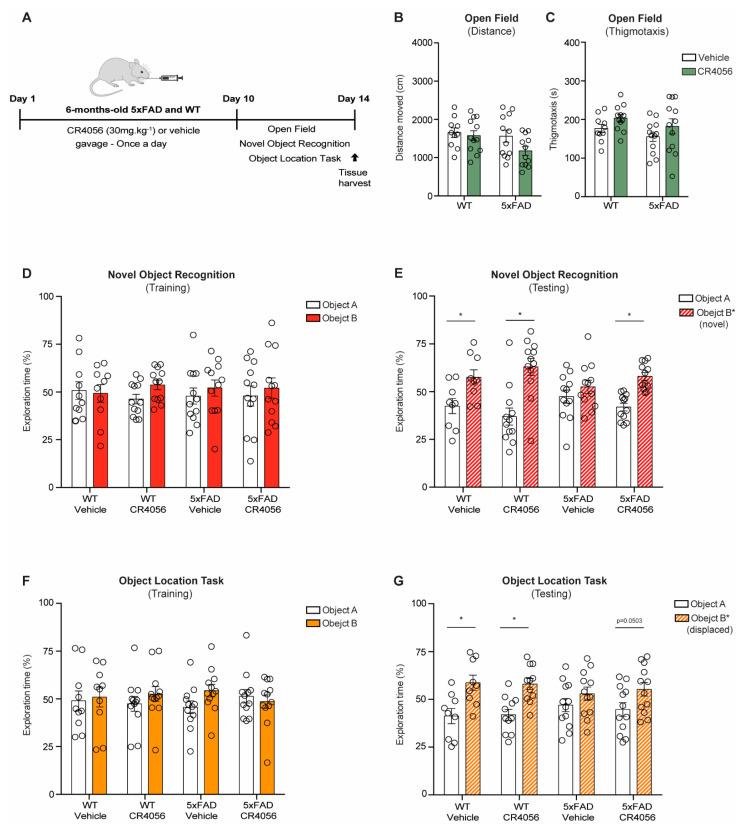
Sub-chronic administration of CR4056 improves recognition memory in 6-month-old 5xFAD mice and not in wild-type (WT) controls. (**A**) Schematic diagram of CR4056 treatment timeline. (**B**,**C**) Measurement of distance moved (**B**) and thigmotaxis (**C**) by open field test (OF) in 5xFAD and WT mice (*n* = 10–12 per group). (**D**) Novel Object Recognition training (NOR) demonstrating the percentage of total exploration of two identical Objects A and B after Vehicle or CR4056 treatment (WT, *n* = 10–12 per group; 5xFAD, *n* = 12 per group). (**E**) Measurement of exploration time during NOR testing, whereby Object B was replaced by a new object (Object B*, WT, *n* = 9–12 per group; 5xFAD, *n* = 12 per group). (**F**) Object Location Task (OLT) training did not show differences in the exploration of Objects A and B between the groups (WT, *n* = 10–12 per group; 5xFAD, *n* = 12 per group). (**G**) Analysis of OLT testing after CR4056 treatment, where Object B was displaced to a different quadrant of the arena (Object B*; WT, *n* = 9–12 per group; 5xFAD, *n* = 12 per group). Columns represent mean ± SEM. Two-way ANOVA with Bonferroni post hoc analysis, * *p* < 0.05.

**Figure 3 ijms-23-07320-f003:**
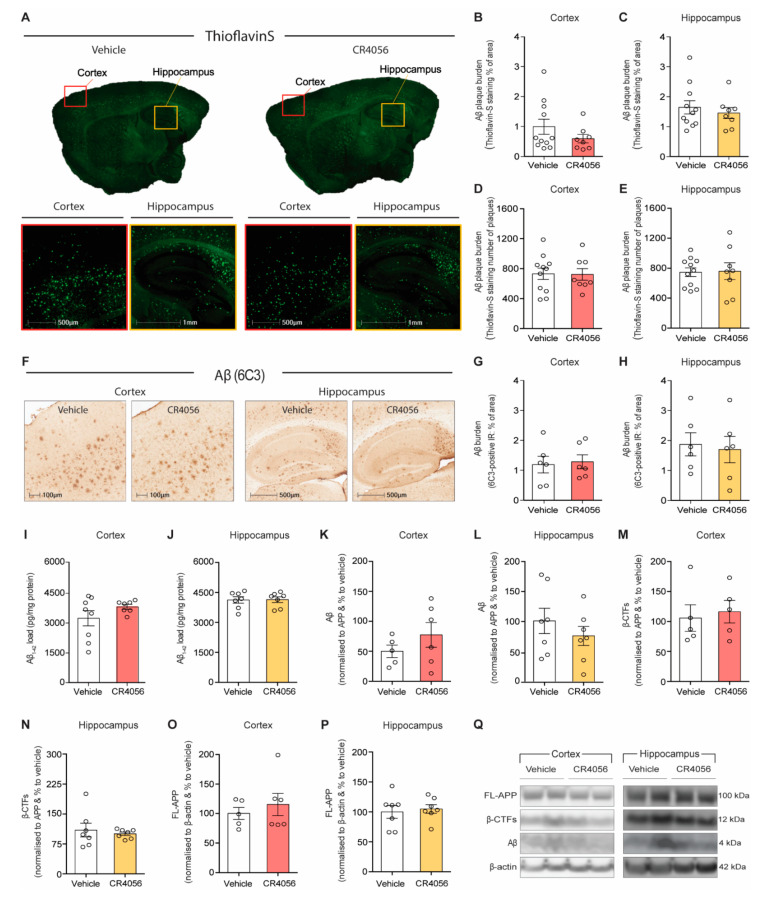
Sub-chronic treatment with CR4056 does not affect Amyloid-β (Aβ) generation and deposition in the cortex and hippocampus of 5xFAD mice. (**A**) Representative images of Thioflavin S staining in sections of 6-month-old 5xFAD mice after treatment with Vehicle or CR4056, respectively. (**B**,**C**) Quantification of percentage of area covered with Aβ plaques by Thioflavin S staining in cortex and hippocampus (*n* = 8–11 per group). (**D**,**E**) Measurement of number of Aβ plaques by Thioflavin S staining in cortex and hippocampus (*n* = 8–11 per group). (**F**) Representative images of anti-Aβ (MOAB-2) immunohistochemistry in cortex and hippocampus of 6-month-old 5xFAD treated with Vehicle or CR4056. (**G**,**H**) Quantification of Aβ load as percentage of area covered with MOAB-2 staining in the cortex and hippocampus, respectively (*n* = 6 per group). (**I**,**J**) Quantification of Aβ_1–42_ by ELISA in cortex and hippocampus (*n* = 7 per group). (**K**–**P**) Quantification of APP processing by Western blot in 5xFAD homogenate from cortex and hippocampus. Quantification of Aβ content in cortex ((**K**), *n* = 5–6 per group) and hippocampus ((**L**), *n* = 7 per group) by Western blot; β-CTFs load in cortex ((**M**), *n* = 5 per group) and hippocampus ((**N**), *n* = 7 per group); and Full-length-APP in cortex ((**O**), *n* = 5–6 per group) and hippocampus ((**P**), *n* = 7 per group). (**Q**) Representative Western blot of APP processing by 6E10 in cortex and hippocampus of 5xFAD mice treated with vehicle or CR4056. Columns represent mean ± SEM.

**Figure 4 ijms-23-07320-f004:**
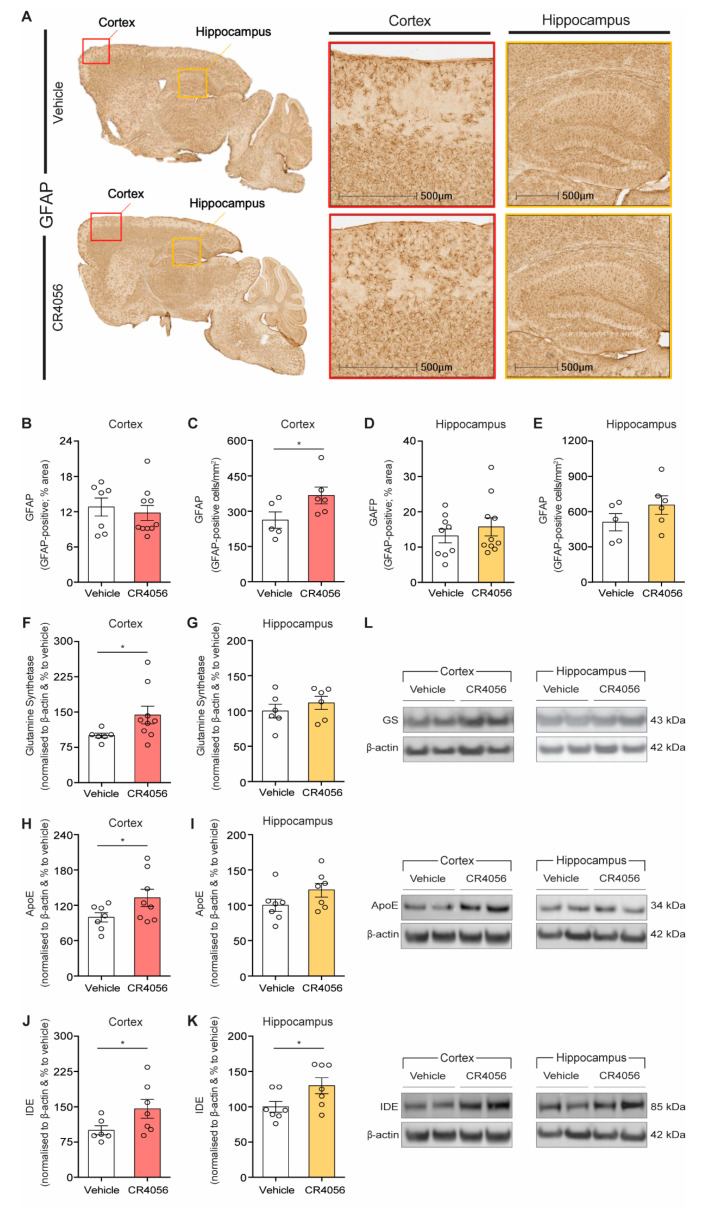
CR4056 increases astrocytic and Aβ clearance markers in 5xFAD mice. (**A**) Representative images of GFAP staining in sections of 6-month-old 5xFAD mice after treatment with vehicle or CR4056. (**B**,**C**) Quantification of percentage of area covered with GFAP-positive staining (*n* = 7–10 per group) and number of GFAP-positive cells (WT *n* = 5–6 per group), respectively, in cortex of 5xFAD mice. (**D**,**E**) Quantification of percentage of area covered with GFAP-positive staining (*n* = 9–10 per group) and number of GFAP-positive cells (*n* = 5–6 per group), respectively, in hippocampus of 5xFAD mice. (**F**,**G**) Quantification of glutamine synthetase (GS) expression and representative western in cortex (WT *n* = 6–8 per group) and hippocampus (*n* = 6 per group), respectively. (**H**,**I**) Quantification of ApoE expression and representative Western blots in homogenates from cortex (*n* = 7–8 per group) and hippocampus (*n* = 7 per group) of 5xFAD mice. (**J**,**K**) Quantification of IDE expression and (**L**) representative Western blots in cortex (*n* = 6–7 per group) and hippocampus (*n* = 7 per group) of 5xFAD mice, respectively. Columns represent mean ± SEM. One-tailed Student’s *t*-test, * *p* < 0.05.

**Figure 5 ijms-23-07320-f005:**
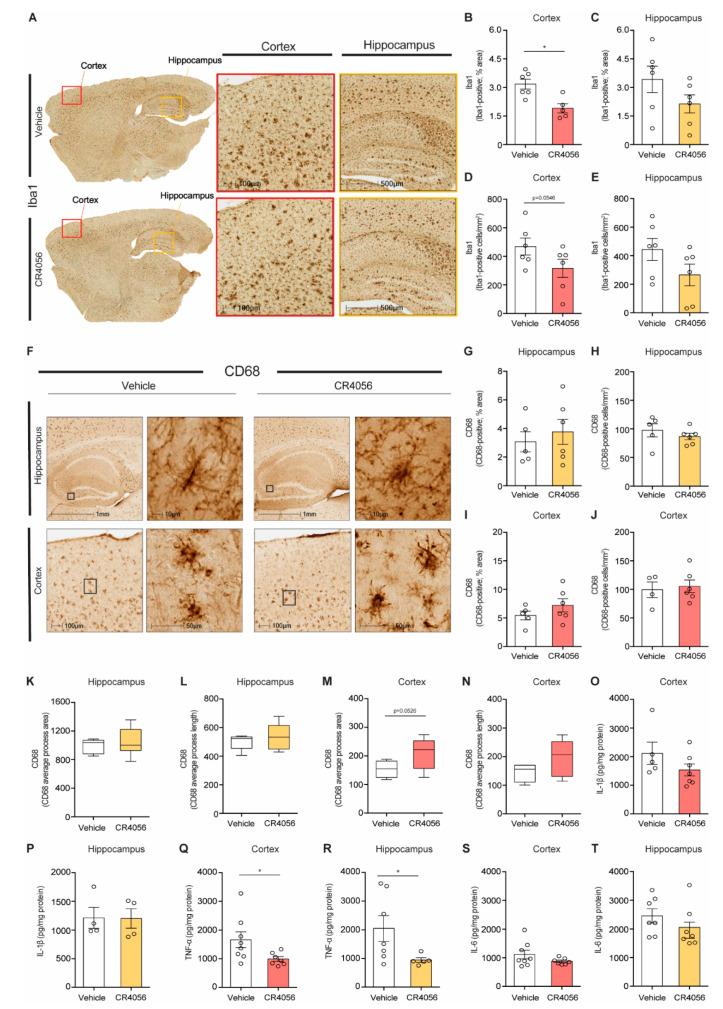
Sub-chronic CR4056 treatment has anti-inflammatory effect in 5xFAD mouse brain. (**A**) Representative images of Iba1 immunostaining in sections from 5xFAD mice after treatment with vehicle or CR4056. (**B**,**C**) Quantification of percentage of area covered with Iba1-positive staining in cortex (*n* = 5–6 per group) and hippocampus (*n* = 6 per group) of 5xFAD mice. (**D**,**E**) Quantification of Iba1-positive cells in cortex and hippocampus of 5xFAD mice, respectively (*n* = 6 per group). (**F**) Representative images of CD68-immunohistochemistry in cortex and hippocampus in sections from 5xFAD mice after with vehicle or CR4056. (**G**,**H**) Quantification of percentage of area covered and number of CD68-positive cells, respectively, in the hippocampus of 5xFAD mice (*n* = 5–6 per group). (**I**,**J**) Quantification of percentage of area covered and number of CD68-positive cells, respectively in the cortex of 5xFAD mice (*n*=5–6 per group). (**K**,**L**) Measurement of process thickness and length of CD68-positive cells in hippocampus (*n* = 5–6 per group) and cortex ((**M**,**N**) *n* = 4–6 per group) of 5xFDA mice. (**O**–**T**) Quantification of expression of cytokines by ELISA in homogenate from cortex and hippocampus of 5xFAD mice after CR4046 treatment. IL-1β ((**O**,**P**), *n* = 4–7 per group), TNF-α ((**Q**,**R**), *n* = 5–8 per group) and IL-6 ((**S**,**T**), *n* = 7–8 per group). Columns represent mean ± SEM. One-tailed Student’s *t*-test, * *p* < 0.05.

**Figure 6 ijms-23-07320-f006:**
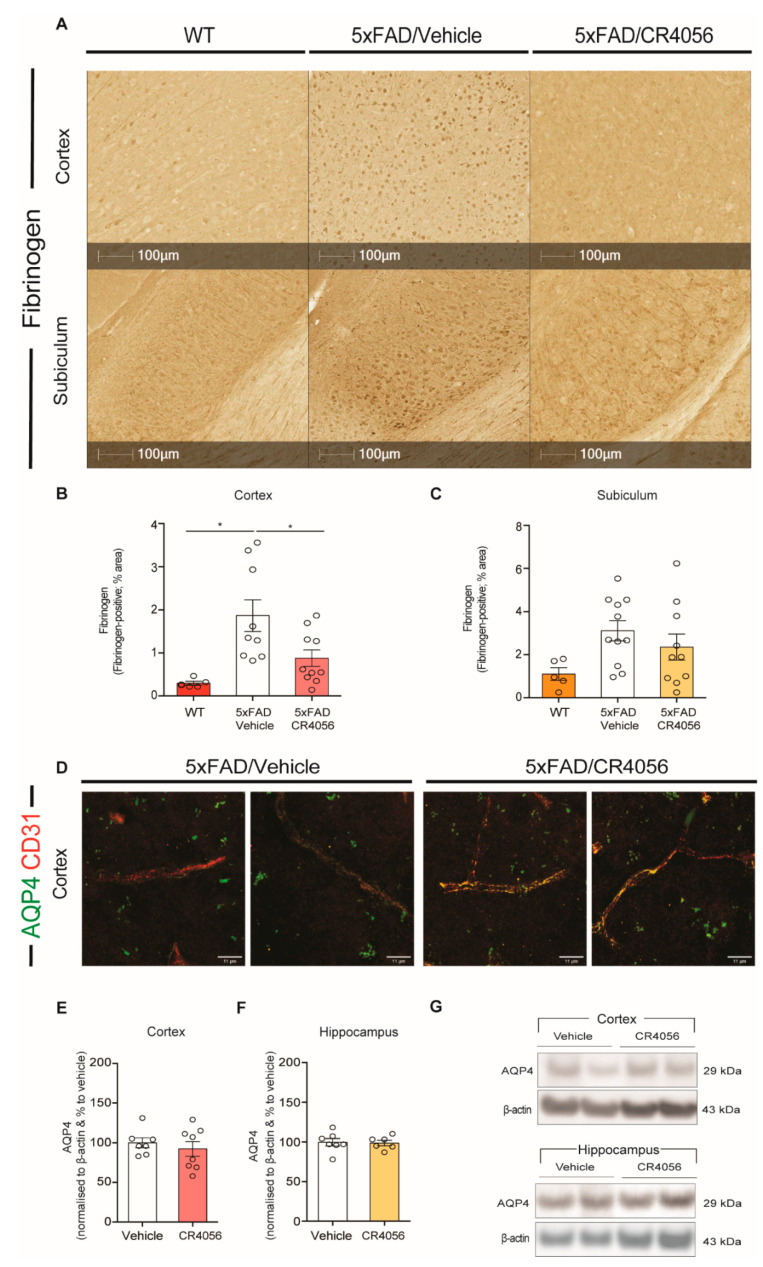
CR4056 restores blood–brain barrier (BBB) dysfunction in 5xFAD mice. (**A**) Representative images of Fibrinogen staining in sections of cortex and subiculum of 6-month-old WT control and 5xFAD mice after treatment with Vehicle or CR5046. (**B**,**C**) Quantification of percentage of area covered with Fibrinogen in cortex and subiculum (*n* = 5–11 per group) of WT-Vehicle and 5xFAD mice treated with vehicle or CR4057. (**D**) Representative images of Aquaporin-4 (AQP4) (green) and CD31 (red) staining showing co-localization in cortical area of sections from 6-month-old 5xFAD mice after CR4056 treatment. Scale bars = 11µm (**E**,**F**) Quantification of AQP4 expression by Western blot in cortex (*n* = 7–8 per group) and hippocampus (*n* = 6–7 per group) of 5xFAD mice. (**G**) Representative Western blot of AQP4 in homogenates from cortex and hippocampus of 5xFAD mice… Columns represent mean ± SEM. One-way ANOVA or *t*-test with Tukey’s post hoc analysis, * *p* < 0.05.

## Data Availability

The data of this study are available from the corresponding author on reasonable request.

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
