# Peer review of "I2-Imidazoline Ligand CR4056 Improves Memory, Increases ApoE Expression and Reduces BBB Leakage in 5xFAD Mice"

_ijms, 2022, doi:10.3390/ijms23137320_

Round 1
Reviewer 1 Report
This manuscript entitled “I2-imidazoline ligand CR4056 improves memory, increases ApoE expression and reduces BBB leakage in 5XFAD mice” describes the therapeutic profiles of I2-imidazoline ligand CR4056 in the 5XFAD model of AD. The study looks interesting because effective drug treatment for AD or other neurodegenerative disorders are in need. The study is comprehensive with solid in vivo biological analysis. All experiments to answer the relevant questions were undertaken. In summary, I recommend this manuscript for acceptance in International Journal of Molecular Sciences.
Minor revisions
1. Authors are suggested to check the manuscript for abbreviations (page 2, line 72), Greek alphabets (Page 2, line 48).
2. Authors are suggested to adjust the figures according to the journal guidelines to avoid unnecessary spaces (page 3,5,7,9 and 11).
Author Response
- Authors are suggested to check the manuscript for abbreviations (page 2, line 72), Greek alphabets (Page 2, line 48).
The abbreviation I2IRs has now been consistently written along the text. The Greek letter is a mistake of the proofs and is correct in the original version.
- Authors are suggested to adjust the figures according to the journal guidelines to avoid unnecessary spaces (page 3,5,7,9 and 11).
We have modified the figures, improved the quality and also provided better representative western blots for figures 4L and 3Q.
Reviewer 2 Report
Mota and coworkers present the neuroprotective effects of the non-opioid analgesic CR4056 in 5XFAD mice. Given the needs for more effective drugs for neurodegenerative diseases such as Alzheimer’s disease, this is always a relevant research topic of health sciences. The manuscript is well written and I would definitely recommend it for publication if authors address the following major and minor concerns:
Major concerns:
1- Introduction: authors should present the chemical structures of all compounds related with the main subject of the manuscript, which include: CR4056, BU224, LSL60101 (aka “Garsevil”) besides all cited drugs for Alzheimer’s disease (donepezil, rivastigmine, galantamine, and memantine). Especial attention should be addressed to the configuration of the stereocenters from galantamine and rivastigmine.
2- Lines 40-41: “Memantine can be administered with an AChEI since they have complementary mechanism of action, for an additive effect.” Please double check the term “additive”. The additive effect is not equal to synergistic effect. Would you mean additive in fact or would be thinking on synergy? If the co-administration of AChE inhibitors and NMDA antagonists has an improved biological response, authors should revise this sentence.
3- All figures are in a poor quality. Authors should improve the quality of all figures before publication.
4- Authors are highlighting the therapeutic potential of CR4056 to be a neuroprotective agent for AD patients. I am wondering about the mutagenic potential and genotoxicity of CR4056. Authors should evaluate the mutagenic profile of CR4056 (e.g., Ames test) to address the safety of the compound for long-term treatments.
5- Lines 258-260: “Interestingly, this drug [CR4056] has been used in clinical trials to treat pain, without side effects, suggesting that it could be used to treat AD patients without having to explore its safety in humans.”. Same concern that I mentioned above. How authors can be sure about the genotoxicity of this compound? Is there any literature that supports compound's safety? If so, authors must cite it and beef up the discussion on this regard.
Minor concerns:
1- Please standardize all acronyms (e.g., AchEi [line 35], AChEI [line 41]; 5XFAD [lines 3, 17, and others], 5xFAD [line 83, 141, and others…]; same for I2R [line 72], I2IR [line 70], I2-IR [line 99]; and others). Authors should double check all the manuscript.
2- Line 48-49: “y-secretase” should be “γ-secretase”.
3- Please define all acronyms the first time they are cited (e.g., NMDA [line 35], BACE1 [line 50], BBB [line 53], GFAP [line 75], APP [line 148], HC [line 241], and others….).
4- Line 76: “(…) ligands have been adapted to have high affinity for I2IRs (…)”. I suggest the replacement of this sentence by a more accurate one such as: “(…) ligands have been designed to have high affinity for I2IRs (…)”.
5- Line 88: “Neuroinflammation,” should be “neuroinflammation,”.
6- Line 171: “Apoliprotein-E” should be “Apolipoprotein-E”.
7- Line 257: “SXFAD” should be “5XFAD,” (please, standardize the acronyms).
8- Line 313: Please replace “ANXA1” by “annexin A1”.
9- Line 389: “H2O2” should be “H2O2”.
10- Line 402: “Xylene” should be “xylene”.
Author Response
Major concerns:
- Introduction: authors should present the chemical structures of all compounds related with the main subject of the manuscript, which include: CR4056, BU224, LSL60101 (aka “Garsevil”) besides all cited drugs for Alzheimer’s disease (donepezil, rivastigmine, galantamine, and memantine). Especial attention should be addressed to the configuration of the stereocenters from galantamine and rivastigmine.
We have now included a new figure (Figure 1) with the chemical structures of the drugs.
- Lines 40-41: “Memantine can be administered with an AChEI since they have complementary mechanism of action, for an additive effect.” Please double check the term “additive”. The additive effect is not equal to synergistic effect. Would you mean additive in fact or would be thinking on synergy? If the co-administration of AChE inhibitors and NMDA antagonists has an improved biological response, authors should revise this sentence.
The study that it is referenced is a systematic review. Perhaps the sentence does not reflect the outcome of the study, whereby combination therapy compared to AChEI monotherapy showed statistically significant effects for cognition and clinical global impression at short term follow-up. The same outcomes have been found in other systematic studies (Guo et al., Brain Behav. 2020 Nov;10(11):e01831). This has been confirmed in studies in animal models, which have shown that the combined use of memantine and the AChEIs can produce greater improvements in measures of memory than either treatment alone.
Both drugs have complementary mechanisms; Memantine addresses dysfunction in glutamatergic transmission, while the AChEIs serve to increase pathologically lowered levels of the neurotransmitter acetylcholine. In addition, preclinical studies have shown that memantine has neuroprotective effects, acting to prevent glutamatergic over-stimulation and the resulting neurotoxicity. Preclinical data have shown how these drugs act via two different, but interconnected, pathological pathways, and that their complementary activity may produce greater effects than either drug individually.
Therefore, it is possible that there is a kind of synergistic effect. However, we have rephrased the sentence, since there is still controversy on this topic.
- All figures are in a poor quality. Authors should improve the quality of all figures before publication.
The quality and resolution of the figures presented is due to the current proof. The original figures submitted had better quality and resolution. We have modified the size and included some better representative images for Thio-S and some westerns (for APP processing in figure 3 and GS in figure 4).
4- Authors are highlighting the therapeutic potential of CR4056 to be a neuroprotective agent for AD patients. I am wondering about the mutagenic potential and genotoxicity of CR4056. Authors should evaluate the mutagenic profile of CR4056 (e.g., Ames test) to address the safety of the compound for long-term treatments.
CR4056 was not mutagenic in the Ames test, both in the absence and in the presence of metabolic activation. No clastogenic activity was found in a preliminary chromosomal aberration test in CHO cells. However, in a GLP test in human lymphocytes, CR4056 significantly increased chromosomal aberrations in the absence of metabolic activation. This finding prompted a thorough investigation of possible genotoxic effects in vivo. Bone marrow micronucleus and unscheduled DNA synthesis (UDS) assays were conducted in rats up to the maximum tolerated dose (i.e. 1000 mg/kg, approximately 170 times the optimal analgesic dose in the same species). The two experiments were performed along with a proof of absorption test aimed at determining whether animals were continuously exposed to CR4056. Both tests were negative despite the high plasma levels of CR4056 at 1 and 4 h after dosing. Moreover, after administration of CR4056 at doses of up to 250 mg/kg in rats, the Comet assay did not show genotoxic responses in the tissues evaluated (stomach, duodenum and liver). Altogether, given the high incidence of spurious positive data in the chromosomal aberration test, these results conceivably exclude a relevant risk of genotoxicity.
5- Lines 258-260: “Interestingly, this drug [CR4056] has been used in clinical trials to treat pain, without side effects, suggesting that it could be used to treat AD patients without having to explore its safety in humans.”. Same concern that I mentioned above. How authors can be sure about the genotoxicity of this compound? Is there any literature that supports compound's safety? If so, authors must cite it and beef up the discussion on this regard.
See previous reply regarding genotoxicity. Regarding safety, the paper Rovati et al. Osteo-522 arthritis Cartilage 2020, 28, 22-30 is cited in the manuscript.
Minor concerns:
1- Please standardize all acronyms (e.g., AchEi [line 35], AChEI [line 41]; 5XFAD [lines 3, 17, and others], 5xFAD [line 83, 141, and others…]; same for I2R [line 72], I2IR [line 70], I2-IR [line 99]; and others). Authors should double check all the manuscript. Thanks; we have done this.
2- Line 48-49: “y-secretase” should be “γ-secretase”. The Greek letter is a mistake of the proofs and is correct in the original version.
3- Please define all acronyms the first time they are cited (e.g., NMDA [line 35], BACE1 [line 50], BBB [line 53], GFAP [line 75], APP [line 148], HC [line 241], and others….). This has been now modified
4- Line 76: “(…) ligands have been adapted to have high affinity for I2IRs (…)”. I suggest the replacement of this sentence by a more accurate one such as: “(…) ligands have been designed to have high affinity for I2IRs (…)”. This has been now modified
5- Line 88: “Neuroinflammation,” should be “neuroinflammation,” This has been now modified
6- Line 171: “Apoliprotein-E” should be “Apolipoprotein-E”. This has been now modified
7- Line 257: “SXFAD” should be “5XFAD,” (please, standardize the acronyms). This has been now modified
8- Line 313: Please replace “ANXA1” by “annexin A1”. This has been now modified
9- Line 389: “H2O2” should be “H2O2”. This has been now modified
10- Line 402: “Xylene” should be “xylene. This has been now modified
Reviewer 3 Report
The manuscript is well written and of interest, due to the lack of therapeutics in an increasing neurodegenerative pathology, such as AD.
Major comment:
-are the effects of CR4056 mediated by the I2 receptor? Could the Authors investigate whether the effects mediated by CR4056 are blocked by a selective I2 receptor antagonist?
Minor comment:
- the bibliography may be improved by citing more recent publications, such as those on Aducanumab failure and the role of microglia in neuroinflammation (acute vs chronic).
Author Response
Major comment
Are the effects of CR4056 mediated by the I2 receptor? Could the Authors investigate whether the effects mediated by CR4056 are blocked by a selective I2 receptor antagonist?
The pharmacology of I2 binding sites is more complex than that of most conventional receptors. Indeed, single agents, such as idazoxan could behave as putative antagonist towards most of I2 triggered effects, still maintaining some “agonist activity” on others, as discussed, for instance, by Vellani et al BJP 2020. Therefore, a conventional agonist/antagonist pharmacology on I2 receptors is difficult to perform. Having said this, CR4056 demonstrated to be specific for I2 receptors, being devoid of activity/affinity towards a large series of cross-targets (Ferrari et al J Pain Res 2011, supplementary material). Consequently, most of its activities are antagonized by the purported I2 antagonist idazoxan but not by other antagonists towards different targets, such as I1, adrenergic α2 and α1, or opiate receptors (Ferrari et al. J Pain Res 2011; Lanza et al, BJP 2014). Based on these findings, also the AD mitigating behavior of CR4056 in AD models should be ascribed to its peculiar I2 ligand profile. In this case CR45056 activity in AD is shared by the other highly selective I2 receptor ligand BU224 (Mirzaei et al. BJP 2021), strongly supporting the specific role of I2 receptors in this AD model.
Minor comment:
- the bibliography may be improved by citing more recent publications, such as those on Aducanumab failure and the role of microglia in neuroinflammation (acute vs chronic). The role of microglia in Neuroinflammation is reviewed in references 10 and 12.
Round 2
Reviewer 2 Report
The manuscript should be considered for publication in present form.